# AttriLens-Mol: Attribute Guided Reinforcement Learning for Molecular Property Prediction with Large Language Models

## Abstract

Large Language Models (LLMs) have shown promise in assisting molecular property prediction tasks but often rely on human-crafted prompts and chain-of-thought templates. While recent advanced large reasoning models like DeepSeek-R1 employ reinforcement learning for an extended "thinking" process, their reasoning can be verbose and lack relevance. We introduce AttriLens-Mol, an attribute-guided reinforcement learning framework for molecular property prediction with LLMs. AttriLens-Mol steers the model's reasoning by using: (1) a format reward encouraging attribute-based structured output, (2) a count reward to avoid enumerating irrelevant attributes, and (3) a rationality reward using advanced LLMs and RDKit to verify the relatedness of the generated attributes. This approach implicitly elicits the model's inherent knowledge of relevant molecular attributes during reasoning, enables making predictions for the molecular property more effectively. Experiments on both in-distribution and out-of-distribution datasets show that, training both 7B-size R1-Distilled-Qwen2.5 and R1-Distilled-LLaMA3.1 models on 4,000 samples with our proposed AttriLens-Mol method significantly boosts the performance, getting comparable or better results than supervised fine-tuning models (Mol-Instructions, ChemDFM, etc.) and advanced models (GPT-3.5, GPT-4o, DeepSeek-V3, DeepSeek-R1, etc.). Further, our extracted attributes for the target property, when used as features for an interpretable decision tree model, yield superior performance compared to attributes generated by prompting LLMs. This shows that AttriLens-Mol effectively elicits more relevant and predictive molecular attributes, leading to enhanced interpretability and performance for property prediction. Code is released at anonymous repository.

## 1 Introduction

The molecular property prediction task aims to determine target properties (such as blood-brain barrier permeability, toxicity, etc.) for given molecular formulas (e.g., SMILES). Traditional methods involve training classification or regression models based on graph neural networks (You et al., 2020; Rong et al., 2020; Fang et al., 2022) or pre-trained models (Ahmad et al., 2022; Ross et al., 2022; Chang & Ye, 2024). However, as shown in Figure 1(a), such models are often only capable of handling specific tasks and perform classification by outputting label probabilities. With the advancement of large language models (LLMs), recent studies attempt to employ them for molecular prediction tasks, including Mol-instructions (Fang et al., 2024), LLaSMol (Yu et al., 2024), and ChemDFM (Zhao et al., 2025b), which can process multi-tasks via user-friendly verbal input and output while also shows promising results.

Despite the success, molecular property prediction based on LLMs still has several limitations. Firstly, the reasoning of LLMs relies on human-written in-context learning (Brown et al., 2020) demonstrations or additional chain-of-thought (CoT; Wei et al., 2022b) prompts. These artificially designed prompts may introduce performance instability, low transferability and poor generalization (Zhao et al., 2021; Min et al., 2022). Secondly, although CoT prompting can improve accuracy to some extent, research indicates that the CoT reasoning process may not correlate with final predictions. For example, Wei et al. (2022b) shows that even flawed reasoning may still yield

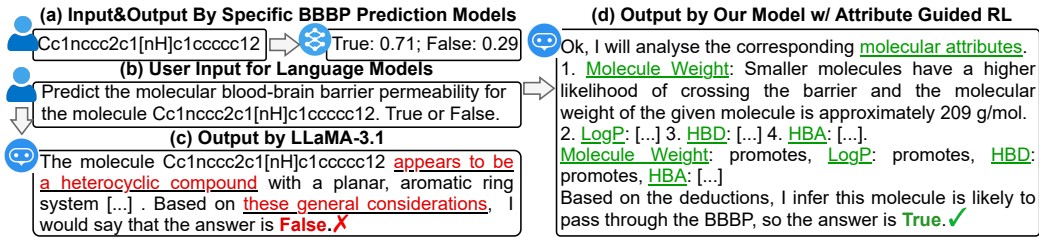

Figure 1: Examples for comparing different methods. (a) Task specific model generates probabilities of labels for given molecule. (b) The verbal input for language models. (c) Incorrect response by LLaMA-3.1. (d) Correct response by our 7B model with attribute guided reinforcement learning, which offers useful attribute information during reasoning and aids property prediction in the final.

correct answers. Therefore, as an example shown in Figure 1(b), CoT-prompting or supervised fine-tuning (Ouyang et al., 2022) methods can still be improved for property prediction.

To enable LLMs to autonomously explore better reasoning paths, inspired by recent large reasoning models via reinforcement learning such as DeepSeek-R1 (Guo et al., 2025), we attempt to train LLMs for molecular property prediction using purely reinforcement learning with verifiable rewards, which does not rely on designing demonstrations or manually-written reasoning steps. Specifically, considering that molecular properties are directly correlated with several key attributes (Zhang et al., 2025; Zheng et al., 2025), we further propose an attribute-guided reinforcement learning method called AttriLens-Mol to make the reasoning process more interpretable and coherent. In addition to the format and correctness based rewards in DeepSeek-R1, AttriLens-Mol introduces attribute-based rewards to better optimize the LLMs and elicit its inherent knowledge and reasoning capabilities related to molecular properties.

The additional attribute-based rewards are in threefold: First, we encourage the model to incorporate property-relevant attributes during its reasoning process, which falls under the rule-based format rewards. Second, to ensure the model focuses on discussing key attributes, we limit the number of relevant attributes to prevent lengthy and unnecessary overthinking. Finally, to further elicit the model's internal knowledge of attributes while reducing potential hallucinations, we externally validate the generated attribute values using both advanced LLMs (GPT-4o and DeepSeek-R1) and cheminformatics RDKit (Landrum et al., 2013) toolkit. Through the optimization of these rewards, the model can generate more relevant, accurate, and concise attribute information related to target properties, thereby enhancing the accuracy of final property predictions.

Experimental results show that, by using different backbones (R1-Distilled-Qwen2.5 and R1-Distilled-LLaMA3.1) and reinforcement learning algorithms (GRPO and DAPO), our 7B-size models show superior or comparable performance against advanced LLMs (GPT-4o, DeepSeek-V3, DeepSeek-R1) as well as pre-trained task-specific models (ChemBERTa, MolBERT, SPMM) on both widely used in-distribution and out-of-distribution datasets. In addition, when combined with interpretable decision tree models, the attributes generated during the reasoning trace via our model provide better performance and interpretability.

## 2 RELATED WORK

**Molecular Property Prediction.** Traditional methods are primarily based on graph neural networks, including GROVER (Rong et al., 2020), GraphCL (You et al., 2020), ChemRL-GEM (Fang et al., 2022), and GeomGCL (Li et al., 2022), which leverage pretrained GNNs to enhance molecular structure representations. S-CGIB (Hoang & Lee, 2025) introduces a subgraph-conditioned bottle-neck to extract graph cores and functional substructures. For Transformer-based models, Chem-BERTa (Ahmad et al., 2022) and MolFormer (Ross et al., 2022) are designed to extract contextual representations from SMILES. SPMM (Chang & Ye, 2024) introduces a cross-attention mechanism to align SMILES sequences with 53 molecular descriptors. Although these task-specific methods have achieved impressive predictive performance, they tend to rely heavily on structural inputs, exhibit limited interpretability via textual interaction.

---

**System**: Your task is to predict the property of the given molecule. You must write your response using the following strict XML format: `<think>` Step-by-step reasoning with consideration on relevant attributes can be calculated using RDKit. For each attribute, provide its estimated value, and explain whether it promotes (improve) or inhibits (not improve) the target property. `</think>`, `<name>` List the attributes you used, each followed by ": promotes" or ": inhibits", separated by commas. For example: attribute A: promotes, attribute B: promotes, attribute C: inhibits. `</name>`, `<answer>` The final answer (e.g., true/false or specific values) based on your overall reasoning. `</answer>`. **User**: The task is {TASK}, the molecule is {SMILES}, and the property to be considered is {TGT}. **Assistant**:

---

Table 1: The template of attribute guided reinforcement learning for molecular property prediction. {TASK} is classification or regression, {SMILES} and {TGT} will be replaced with the specific molecule and property, respectively.

**Property Prediction in LLMs.** Mol-Instructions (Fang et al., 2024), LLaSMol (Yu et al., 2024), ChemDFM (Zhao et al., 2025b) and GeLLM$^3$O (Dey et al., 2025) construct high-quality molecular datasets covering multiple tasks and employ instruction tuning (Wei et al., 2022a) to enhance the models' ability to understand and reason about molecular structures, properties, reactions, and their interrelations. However, different from reinforcement learning based methods, these works rely on the integration of external knowledge to enhance model capabilities, rather than stimulating the models' intrinsic reasoning potential.

**Enhancing Reasoning via Reinforcement Learning.** DeepSeek-Math (Shao et al., 2024) and DeepSeek-R1 (Guo et al., 2025) stand as seminal works that pioneered leveraging reinforcement learning to advance reasoning abilities of LLMs in diverse domains. For examples, StepCoder (Dou et al., 2024) employs reinforcement learning to progressively optimize the decomposition and generation of complex programming tasks. MT-R1-Zero (Feng et al., 2025) focuses on machine translation tasks by designing a semantic reward mechanism that guides the model to learn translation strategies. To our knowledge, we are the first to propose using several attribute-related rewards for better molecular property prediction via a pure reinforcement learning method.

## 3 METHOD

We first design a template (Section 3.1) and apply two rule-based format and correctness rewards to facilitate an accurate thinking path for molecular property prediction (Section 3.2). We also incorporate attribute-oriented count and rational rewards to generate related attributes and for better prediction (Section 3.3). Finally, we can apply existing algorithms for RL training (Section 3.4).

### 3.1 ATTRIBUTE-BASED STRUCTURED TEMPLATE

To guide the model to adhere instructions for molecular property prediction, we design a template for the training process of our AttriLens-Mol, as shown in Table 1.

Building upon DeepSeek-R1 (Guo et al., 2025) with the thinking process enclosed in the `<think></think>` tags and answer tags `<answer></answer>`, we additionally incorporate requirements for in-depth discussion of relevant attributes (e.g., "reasoning with consideration on relevant attributes") and summarize how these attributes affect the target property (e.g., promote or inhibit) within the `<name></name>` tags. Thus we can incorporate relevant attributes in the structured response and lead to the final prediction answer accordingly, facilitating attribute guided molecular property prediction.

### 3.2 TASK-AGNOSTIC RULE-BASED REWARDS

The overall rewards to be optimized in our AttriLens-Mol are shown in Figure 2. We first follow DeepSeek-R1 which employs rule-based task-agnostic rewards in two aspects: whether the model's response contains specific formatting information and whether the correct answer is derived accordingly. We expect this approach to perform well in tasks with verifiable outcomes (e.g., whether a molecule exhibits toxicity) while providing user-friendly textual explanations.

Figure 2: Illustration of our AttriLens-Mol reinforcement learning. $\mathcal{R}^{\text{format}}$ and $\mathcal{R}^{\text{correct}}$ are rule-based general rewards (Section 3.2), $\mathcal{R}^{\text{count}}$ and $\mathcal{R}^{\text{rational}}$ are our attribute-based rewards for molecular property prediction (Section 3.3), respectively.

**Format-based Reward.** The format-based reward $\mathcal{R}^{\text{format}}$ encourages the model to output a reasoning process within `<think></think>` tags, generate relevant molecular attributes considered between `<name></name>` tags, and finally include the answer (e.g., true or false for binary classification) within `<answer></answer>` tags. $\mathcal{R}^{\text{format}}$ is calculated by:

$$\mathcal{R}^{\text{format}} = \begin{cases} 1, & \text{if the format is correct,} \\ -2, & \text{if the format is incorrect.} \end{cases} \tag{1}$$

**Correctness-based Reward.** The correctness-based reward $\mathcal{R}^{\text{correct}}$ encourages the model to output the accurate predictions for training data and is calculated by:

$$\mathcal{R}^{\text{correct}} = \begin{cases} 2, & \text{if the answer is correct,} \\ 0, & \text{if the answer is incorrect,} \end{cases} \tag{2}$$

where the final answer can be extracted within the designed `<answer></answer>` tags in response, and the labels from training set can be used for correctness verification. Note that we apply the widely used two-way classification training samples (e.g., BBBP), thus $\mathcal{R}_{\text{correct}}$ can be accurately computed according to whether the response is true or false, and we find that this classification-based reward can also enhance performance on out-of-distribution ESOL and FreeSolv datasets which are regression tasks (Section 4.2).

## 3.3 TASK-RELATED ATTRIBUTE-ORIENTED REWARDS

**Count-based Reward.** The reasons for limiting the number of attributes are two-fold: 1) Existing research show that large reasoning models tends to be overthinking (Chen et al., 2024; Zhao et al., 2025a), where reasoning chains progressively lengthen during training. This results in increased computational costs during inference and could also involves unrelated contents. Our approach directs models to focus on deriving critical attributes rather than exhaustively enumerating irrelevant ones. 2) Systematic studies on the number of attributes relevant to molecular property prediction tasks remain challenge. Empirical works such as MPCD (Yang et al., 2024b) adopts a multi-task learning approach by jointly training on samples from 12 different attributes datasets to enhance the model's prediction capability. AutoMolCo (Zhang et al., 2025) utilizes LLMs for generating and labeling molecular concepts, where the number of selected concepts typically falls between 7 and 9. Based on these consideration, we adopt a similar range 3~10 for keeping the most related attributes via count-based reward. We also compare different settings in ablation study in Section 4.3.

As illustrated in Table 1, the `<name></name>` tags are designated to conclude the names of molecular attributes considered during reasoning. The count-based reward $\mathcal{R}^{\text{count}}$ is calculated by:

$$\mathcal{R}^{\text{count}} = \begin{cases} 0, & \text{if } n_{\text{att}} \in [3, 10], \\ -1, & \text{otherwise,} \end{cases} \tag{3}$$

where $n_{\text{att}}$ denotes the total number of attributes in the `<name></name>` tags during reasoning.

**Rationality-based Reward.** In aforementioned reward designs, the model could adequately adhere to attribute-related formats and quantities, and might even provide correct answers. However, the connection between these attributes and the final answer could still remain tenuous. To address this

Figure 3: Illustration of calculating the attribute rationality $\mathcal{R}^{\text{rational}}$ using external advanced LLMs and cheminformatics toolkit for each single attribute value alignment.

issue, we propose a rationality-based reward for assessing the quality and coherence of the attributes generated during the reasoning process, as shown in Figure 3.

The output of "promotes" in `<name></name>` tags indicates that the attribute contributes positively to the target property, while "inhibits" suggests a negative contribution. We adopt a fuzzy matching strategy to map the extracted attributes to standardized attribute names supported by RD-Kit (Landrum et al., 2013), and subsequently calculate their precise values for the given molecule. In particular, although the performance of LLMs in predicting molecular properties remains to be improved, they can still provide relatively reliable molecular attribute information by leveraging parametric knowledge, as shown in LLM4SD (Zheng et al., 2025). Thereby we pre-define advantageous value ranges for all 53 RDKit-computable molecular attributes by prompting advanced LLMs such as GPT-4o and DeepSeek-R1 (we compare these two LLMs in Section 5.3). The rationality aligns (or not) if an attribute's value falls within (or out of) its advantageous range and the prediction is "promotes" (or "inhibits"). This enables us to assess verbally how the model identifies the impact of considered attribute for the final response.

Formally, we first extract the rationality by model $r = r_1, r_2, ..., r_{|\mathcal{A}|}$ for $|\mathcal{A}|$ generated attributes, where $r_i$ is 1/0 if the response is "promotes" or "inhibits" accordingly. Then we calculate the rationality by LLM and RDKit $\hat{r} = \hat{r}_1, \hat{r}_2, ..., \hat{r}_{|\mathcal{A}|}$, where $\hat{r}_i$ is 1/0 if the corresponding value is within (or out of) the range by LLMs. The rationality reward $\mathcal{R}^{\text{rational}}$ is calculated by:

$$\mathcal{R}^{\text{rational}} = \frac{1}{|\mathcal{A}|} \sum_{i=1}^{|\mathcal{A}|} \mathbb{1}\{r_i = \hat{r}_i\}, \tag{4}$$

where the range of value is 0~1, serving as a fine-grained indicator of the model's consistency in assessing the impact of attributes for the target property. Note that we use this reward to validate the coherence of the generated attributes, not encouraging models to mimic the reasoning trace of advanced GPT-4o or DeepSeek-R1 models.

## 3.4 REINFORCEMENT LEARNING ALGORITHM

We employ the group relative policy optimization (GRPO; Shao et al., 2024) for training. At each step, given a query $q$, a group of candidate outputs $\{o_i\}_{i=1}^G$ is sampled from the current policy $\pi_{\theta_{\text{old}}}$. The advantage $A_i$ of each candidate is computed by normalizing its reward $r_i$ within the group:

$$A_i = \frac{r_i - \text{mean}(\{r_j\})}{\text{std}(\{r_j\})}. \tag{5}$$

GRPO updates the policy $\pi_\theta$ by maximizing the clipped surrogate objective with a KL penalty:

$$\mathcal{J}_{\text{GRPO}}(\theta) = \mathbb{E}_{q \sim \mathcal{P}(Q), \{o_i\}_{i=1}^G \sim \pi_{\theta_{\text{old}}}(\cdot|q)} \left[ \frac{1}{G} \sum_{i=1}^G \min \left( \frac{\pi_\theta(o_i|q)}{\pi_{\theta_{\text{old}}}(o_i|q)} A_i, \text{clip}\left( \frac{\pi_\theta(o_i|q)}{\pi_{\theta_{\text{old}}}(o_i|q)}, 1-\epsilon, 1+\epsilon \right) A_i \right) - \beta\, D_{\text{KL}}(\pi_\theta \| \pi_{\text{ref}}) \right] \tag{6}$$

where the KL divergence is approximated by:

$$D_{\text{KL}}(\pi_\theta \| \pi_{\text{ref}}) = \frac{\pi_{\text{ref}}(o_i|q)}{\pi_\theta(o_i|q)} - \log \frac{\pi_{\text{ref}}(o_i|q)}{\pi_\theta(o_i|q)} - 1, \tag{7}$$

| Models | Reasoning Trace | Attr. Interp. | In-Distribution | | | Out-of-Distribution | | | AVG$_4\uparrow$ | AVG$_2\downarrow$ |
|---|---|---|---|---|---|---|---|---|---|---|
| | | | BBBP Acc.↑ | BACE Acc.↑ | ClinTox Acc.↑ | SIDER Acc.↑ | FreeSolv RMSE↓ | ESOL RMSE↓ | | |
| *Pre-Trained Task Specific Models* | | | | | | | | | | |
| ChemBERTa (Ahmad et al., 2022) | ✗ | ✗ | 60.3 | 69.7 | 93.9 | - | - | - | - | - |
| MolBERT (Xia et al., 2023) | ✗ | ✗ | 63.2 | 71.7 | 94.6 | - | - | - | - | - |
| SPMM (Chang & Ye, 2024) | ✗ | ✗ | 59.3 | 66.5 | 93.5 | - | - | - | - | - |
| *Large Language Models w/o Tuning (>100B)* | | | | | | | | | | |
| GPT-3.5-turbo (OpenAI, 2023b) | ✗ | ✗ | 51.0 | 49.3 | 50.0 | 46.2 | 5.55 | 2.04 | 49.1 | 3.80 |
| GPT-3.5-turbo (OpenAI, 2023b) +CoT | ✓ | ✗ | 55.4 | 54.6 | 50.7 | 49.7 | 5.79 | 3.37 | 52.6 | 4.58 |
| GPT-4o (OpenAI, 2023a) | ✗ | ✗ | 62.8 | 50.0 | 50.0 | 52.5 | 6.09 | 8.17 | 53.8 | 7.13 |
| GPT-4o (OpenAI, 2023a) +CoT | ✓ | ✗ | 59.8 | 52.6 | 52.7 | 53.9 | 7.17 | 3.05 | 54.8 | 5.11 |
| DeepSeek-V3 (Liu et al., 2024) | ✗ | ✗ | 46.6 | 44.1 | 39.9 | 55.9 | 6.01 | 3.22 | 46.6 | 4.61 |
| DeepSeek-V3 (Liu et al., 2024) +CoT | ✓ | ✗ | 57.8 | 52.0 | 35.1 | 57.3 | 4.36 | 3.63 | 50.6 | 4.00 |
| DeepSeek-R1 (Guo et al., 2025) | ✓ | ✗ | 66.7 | 57.8 | 54.1 | 55.9 | 4.55 | 4.14 | 58.6 | 4.35 |
| *Large Language Models w/o Tuning (<10B)* | | | | | | | | | | |
| LLaMA3.1-8B (Dubey et al., 2024) | ✗ | ✗ | 45.6 | 16.5 | 19.6 | 52.5 | 76.22 | 86.41 | 33.7 | 81.32 |
| LLaMA3.1-8B (Dubey et al., 2024) +CoT | ✓ | ✗ | 54.9 | 57.5 | 29.0 | 54.1 | 18.04 | 19.43 | 48.8 | 18.74 |
| Qwen2.5-7B (Yang et al., 2024a) | ✗ | ✗ | 49.0 | 41.5 | 32.4 | 48.3 | 127.77 | 21.75 | 42.8 | 74.76 |
| Qwen2.5-7B (Yang et al., 2024a) +CoT | ✓ | ✗ | 52.5 | 59.2 | 29.7 | 53.9 | 44.90 | 12.90 | 48.9 | 28.90 |
| Qwen3-8B (Yang et al., 2025) | ✗ | ✗ | 50.5 | 43.4 | 34.5 | 53.1 | 63.71 | 19.77 | 45.4 | 41.74 |
| Qwen3-8B (Yang et al., 2025) +CoT | ✓ | ✗ | 53.2 | 57.2 | 33.8 | 51.8 | 23.42 | 12.22 | 49.0 | 17.82 |
| *Large Language Models w/ Supervised Fine-Tuning (<10B)* | | | | | | | | | | |
| Mol-Instructions-8B (Fang et al., 2024) | ✗ | ✗ | 54.4 | 51.3 | 53.4 | 41.7 | 19.15 | 17.64 | 50.2 | 18.40 |
| ChemDFM-8B (Zhao et al., 2025b) | ✗ | ✗ | 56.9 | 51.9 | 58.6 | 42.0 | 21.35 | 23.25 | 52.4 | 22.30 |
| LLaMA3.1-8B (Dubey et al., 2024) | ✗ | ✗ | 54.4 | 60.5 | 89.9 | 45.3 | 16.06 | 14.42 | 62.5 | 15.24 |
| Qwen2.5-7B (Yang et al., 2024a) | ✗ | ✗ | 52.5 | 55.3 | 89.8 | 55.2 | 13.15 | 57.00 | 63.2 | 35.10 |
| Qwen3-8B (Yang et al., 2025) | ✗ | ✗ | 52.0 | 58.6 | 90.1 | 48.2 | 54.89 | 13.15 | 62.2 | 34.02 |
| *Large Language Models w/ Reinforcement Learning (<10B)* | | | | | | | | | | |
| R1-Distilled-Qwen2.5-7B (Guo et al., 2025) | ✓ | ✗ | 51.8 | 49.1 | 34.0 | 48.9 | 9.36 | 7.96 | 46.0 | 8.66 |
| R1-Distilled-LLaMA3.1-8B (Guo et al., 2025) | ✓ | ✗ | 50.6 | 55.7 | 33.6 | 52.4 | 47.53 | 5.06 | 48.1 | 26.30 |
| **R1-Distilled-Qwen2.5-7B (Ours, GRPO)** | ✓ | ✓ | **58.1** | 60.8 | 88.5 | **56.9** | 11.12 | **1.84** | 66.1 | 6.48 |
| **R1-Distilled-LLaMA3.1-8B (Ours, GRPO)** | ✓ | ✓ | 57.2 | **67.6** | 87.8 | **58.9** | 18.44 | 3.26 | **67.9** | 10.85 |
| **R1-Distilled-Qwen2.5-7B (Ours, DAPO)** | ✓ | ✓ | 56.9 | 62.6 | 91.2 | 54.4 | **7.82** | 2.67 | 66.3 | **5.25** |
| **R1-Distilled-LLaMA3.1-8B (Ours, DAPO)** | ✓ | ✓ | 53.4 | 58.6 | **93.7** | 52.9 | 15.48 | 7.87 | 64.7 | 11.68 |

Table 2: **Attr. Interp.** denotes attribute-based reasoning process. **AVG**$_4$ and **AVG**$_2$ are the averaged results of the first four classification and last two regression tasks. Among large language models with parameters less than 10B, the best result is **in bold** and the second-best result is underlined.

where $\pi_{\text{ref}}$ is typically the initial policy.

In addition, we also use decoupled clipping and dynamic sampling policy optimization (DAPO; Yu et al., 2025) as a variant of GRPO with different clipping and sampling strategy to validate our AttriLens-Mol method. In practice, we observe that the performance of AttriLens-Mol by these two algorithms is similar, both yielding positive results across different datasets.

## 4 EXPERIMENTS

### 4.1 EXPERIMENTAL SETUP

**Baselines.** For task-specific pre-trained models, we use ChemBERTa (Ahmad et al., 2022), Mol-BERT (Xia et al., 2023), and SPMM (Chang & Ye, 2024). For large language models, we use advanced models including ChatGPT-3.5 (OpenAI, 2023b), GPT-4o (OpenAI, 2023a), DeepSeek-V3 (Liu et al., 2024), DeepSeek-R1 (Guo et al., 2025), and also popular open-source models including LLaMA 3.1 (Dubey et al., 2024), Qwen2.5 (Yang et al., 2024a), and Qwen3 (Yang et al., 2025). Based on these models, we apply direct prompting, general chain-of-thought (CoT; Wei et al., 2022b) prompting, supervised fine-tuning (SFT; Ouyang et al., 2022), and reinforcement learning with verifiable reward (RLVR; Shao et al., 2024) for comparison. We also compare attribute-oriented CoT prompting in Section 5.2.

**Datasets.** For policy optimization, we use training sets from three widely used BBBP, BACE, and ClinTox in MoleculeNet (Wu et al., 2018). We apply the prompt template in Table 1 to construct 4,023 structured training samples, serving as guidance for enabling the model to learn autonomous attribute-guided reasoning during property prediction.

| Models | BBBP↑ | BACE↑ | ClinTox↑ | SIDER↑ | FreeSolv↓ | ESOL↓ | Avg$_4$↑ | Avg$_2$↓ |
|---|---|---|---|---|---|---|---|---|
| | | | *R1-Distilled-Qwen2.5* | | | | | |
| Base Model | 51.8 | 49.1 | 34.0 | 48.9 | 9.36 | 7.96 | 45.95 | 8.66 |
| | | | *R1-Distilled-Qwen2.5 (Ours, GRPO)* | | | | | |
| Full Model | **58.1** | **60.8** | 88.5 | **56.9** | 11.12 | **1.84** | **66.08** | **6.48** |
| *w/o* $\mathcal{R}^{\text{rational}}$ | 55.7 (-2.4) | 52.4 (-8.4) | 81.7 (-6.8) | 53.8 (-3.1) | 12.32 (+1.20) | 7.16 (+5.32) | 60.90 (-5.18) | 9.74 (+3.26) |
| *w/o* $\mathcal{R}^{\text{count}}$ | 50.0 (-8.1) | 42.6 (-18.2) | 82.1 (-6.4) | 49.7 (-7.2) | 11.84 (+0.72) | 5.98 (+4.14) | 56.10 (-9.98) | 8.91 (+2.43) |
| *w/o both* | 51.0 (-7.1) | 51.4 (-9.4) | 78.9 (-9.6) | 47.6 (-9.3) | 15.79 (+4.67) | 7.80 (+5.96) | 57.23 (-8.85) | 11.80 (+5.32) |
| *w/* $\mathcal{R}^{\text{count}}, n_{\text{att}} \in [3, 15]$ | 57.9 (-0.2) | 57.2 (-3.6) | **89.1** (+0.6) | 53.5 (-3.4) | **10.79** (-0.33) | 3.07 (+1.23) | 64.43 (-1.65) | 6.93 (+0.45) |
| *w/* $\mathcal{R}^{\text{count}}, n_{\text{att}} \in [3, 20]$ | 54.9 (-3.2) | 55.0 (-5.8) | 87.9 (-0.6) | 52.1 (-4.8) | 13.53 (+2.41) | 3.23 (+1.39) | 62.48 (-3.60) | 8.38 (+1.90) |
| | | | *R1-Distilled-Qwen2.5 (Ours, DAPO)* | | | | | |
| Full Model | 56.9 | 62.6 | 91.2 | 54.4 | 7.82 | 2.67 | 66.28 | 5.25 |
| *w/o* $\mathcal{R}^{\text{rational}}$ | 54.6 (-2.3) | 58.8 (-3.8) | 86.3 (-4.9) | 51.9 (-2.5) | 10.51 (+2.69) | 6.48 (+3.81) | 62.90 (-3.38) | 8.50 (+3.25) |
| *w/o* $\mathcal{R}^{\text{count}}$ | 55.6 (-1.3) | 30.9 (-31.7) | 81.2 (-10.0) | 49.6 (-4.8) | 8.26 (+0.44) | 4.72 (+2.05) | 54.33 (-11.95) | 6.49 (+1.24) |
| *w/o both* | 51.7 (-5.2) | 53.3 (-9.3) | 83.5 (-7.7) | 48.0 (-6.4) | 11.83 (+4.01) | 9.96 (+7.29) | 59.13 (-7.15) | 10.90 (+5.65) |
| *w/* $\mathcal{R}^{\text{count}}, n_{\text{att}} \in [3, 15]$ | 57.7 (+0.8) | 59.6 (-3.0) | 90.4 (-0.8) | 52.4 (-2.0) | 10.13 (+2.31) | 3.36 (+0.69) | 65.03 (-1.25) | 6.75 (+1.50) |
| *w/* $\mathcal{R}^{\text{count}}, n_{\text{att}} \in [3, 20]$ | 55.1 (-1.8) | 59.1 (-3.5) | 83.2 (-8.0) | 53.6 (-0.8) | 9.75 (+1.93) | 3.59 (+0.92) | 62.75 (-3.53) | 6.67 (+1.42) |

Table 3: Ablation study on $\mathcal{R}^{\text{rational}}$ and $\mathcal{R}^{\text{count}}$. The values in parentheses show the differences.

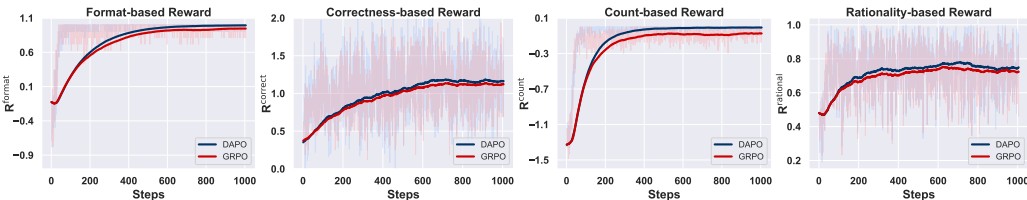

Figure 4: The curves of different rewards during GRPO and DAPO training.

For in-distribution evaluation, we use the test set of BBBP, BACE, and ClinTox. For out-of-distribution (OOD) evaluation, we use the test set of SIDER, and also two regression tasks including FreeSolv and ESOL from MoleculeNet. The statistics of the datasets used are show in Appendix A and the training details are shown in Appendix B.

## 4.2 MAIN RESULTS

The main results are shown in Table 2 and we summarize the key findings as follows:

**Our results are the best among 7B-size models and approximate task-specific or surpasses advanced LLMs.** We achieve 64.7~67.9 Avg$_4$ score on classification tasks and 5.25~11.68 Avg$_2$ score on regression tasks, outperforming 7B-size foundation models (LLaMA and Qwen), molecular task-tuned models (Mol-Instructions and ChemDFM), and reasoning models (R1-Distilled-Qwen2.5 and R1-Distilled-LLaMA3.1), while demonstrating superior or comparable performance to both 100B+ LLMs (49.1~58.6 Avg$_4$ & 3.80~7.13 Avg$_2$), as well as pre-trained task-specific models which can not handle out-of-distribution tasks.

**Compared to CoT, RL-based methods can further elicit the reasoning ability and improve performance.** We observe measurable performance gains when applying CoT prompting to LLMs. For instance, Qwen2.5's Avg$_4$ score increases from 42.9 to 48.9, and Avg$_2$ score decreases from 74.76 to 28.90 after CoT prompting. However, this approach still falls short of RL-trained models. This indicates that, compared to external prompting techniques, RL-based methods elicit intrinsic model capabilities for better self-exploration and reasoning.

**Our method demonstrates out-of-distribution generalization, even on challenging regression tasks.** Our method also improves the performance on OOD tasks SIDER, FreeSolv, and ESOL. Particularly for precision-demanding attribute-value regression tasks, which are known to challenge for LLMs. Our method delivers competitive results despite no specialized reward design for attribute value prediction. For instance, Qwen2.5 trained with DAPO achieves a notable 5.25 Avg$_2$ score. The supervised fine-tuning models, in contrast, does not improve stably for OOD tasks, which is consistent with the findings of Chu et al. (2025) that SFT tends to memorize while RL can achieve generalizable knowledge.

| Methods | BBBP↑ | BACE↑ | ClinTox↑ | SIDER↑ | FreeSolv↓ | ESOL↓ | AvG$_4$↑ | AvG$_2$↓ |
|---|---|---|---|---|---|---|---|---|
| | | | | *R1-Distilled-Qwen2.5* | | | | |
| Attribute-guided CoT | 55.2 | 59.6 | 88.4 | 53.6 | 8.52 | 4.65 | 64.20 | 6.59 |
| Attribute-guided RL (Ours) | **56.9** (+1.7) | **62.6** (+2.9) | **91.2** ( +2.8) | **54.4** (+0.8) | **7.82** (-0.70) | **2.67** (-1.98) | **66.28** (+2.08) | **5.25** (-1.34) |
| | | | | *R1-Distilled-LLaMA3.1* | | | | |
| Attribute-guided CoT | 52.5 | 55.8 | 90.4 | 50.3 | 20.72 | 9.19 | 62.25 | 14.96 |
| Attribute-guided RL (Ours) | **53.4** (+0.9) | **58.6** (+2.8) | **93.7** ( +3.3) | **52.9** (+2.6) | **15.48** (-4.52) | **7.87** (-1.32) | **64.65** (+2.40) | **11.68** (-3.28) |

Table 4: Comparison of attribute-guided CoT and RL. Values in parentheses show our improvement.

| Models | BBBP↑ | BACE↑ | ClinTox↑ | SIDER↑ | FreeSolv↓ | ESOL↓ | AvG$_4$↑ | AvG$_2$↓ |
|---|---|---|---|---|---|---|---|---|
| | | | | *R1-Distilled-Qwen2.5* | | | | |
| *w/o* Attribute-guided RL | 51.8 | 49.1 | 34.0 | 48.9 | 9.36 | 7.96 | 45.95 | 8.66 |
| | | | *R1-Distilled-Qwen2.5 (Ours, DAPO)* | | | | | |
| *w/* guidance by GPT-4o | 56.9 | **62.6** | **91.2** | **54.4** | **7.82** | **2.67** | **66.27** | **5.25** |
| *w/* guidance by DeepSeek-R1 | **57.3** | 60.6 | 90.7 | 53.5 | 8.77 | 2.96 | 65.52 | 5.86 |

Table 5: Comparison of GPT-4o and DeepSeek-R1 as guidance for rationality reward $\mathcal{R}^{\text{rational}}$.

**Our method demonstrates stable results across multiple models and RL algorithms.** Our method demonstrates consistent and stable performance improvements across both R1-Distilled-Qwen2.5 and R1-Distilled-LLaMA3.1 models, as well as different RL algorithms including GRPO and DAPO, confirming its effectiveness.

### 4.3 ABLATION STUDY ON ATTRIBUTE-RELATED REWARDS

The results on ablation study of count-based reward $\mathcal{R}^{\text{count}}$ and rationality-based reward $\mathcal{R}^{\text{rational}}$ are shown in Table 3. We observe varying degrees of performance degradation by removing either $\mathcal{R}^{\text{count}}$ or $\mathcal{R}^{\text{rational}}$ using different algorithms. Notably, the removal of $\mathcal{R}^{\text{count}}$ led to the most significant drop on BACE, even falling below R1-Distilled-Qwen2.5. This phenomenon likely stems from the stronger correlation between the BACE task and the attribute selection and reasoning process, where removing $\mathcal{R}^{\text{count}}$ or setting larger number of attributes leads to more unrelated attributes and incorrect responses, showing the crucial role of $\mathcal{R}^{\text{count}}$ in maintaining stable and effective reasoning.

## 5 ANALYSES AND DISCUSSIONS

### 5.1 PROCESS OF REWARD MODELING

The optimization process of different rewards are shown in Figure 4. For different algorithms, DAPO and GRPO show similar behaviors, where DAPO shows slightly better reward values. For different rewards $\mathcal{R}^{\text{format}}$ and $\mathcal{R}^{\text{count}}$ converge relatively quickly, and the variance of their reward values is small, indicating that the format and count can be learned relatively quickly by the model. For $\mathcal{R}^{\text{correct}}$ and $\mathcal{R}^{\text{rational}}$, since the accuracy and the relevance between attributes and the target property are more difficult to learn, the reward values are relatively unstable. Nevertheless, the obtained rewards are gradually increasing, indicating that the model is effectively learning attribute-based reasoning. Overall, after about 1000 steps, all rewards can basically reach a convergent state, achieving the result of the model possessing good reasoning ability on the given training data.

### 5.2 COMPARISON BETWEEN ATTRIBUTE COT AND RL

We further compare attribute-based CoT, where we prompting LLMs to reason according to attribute-related causality. The results are shown in Table 4. We find that our RL-based method consistently leads to better performance, showing that, compared with prompting-based method, our RL training can better elicit the attribute-guided reasoning ability.

### 5.3 RANGE OF ATTRIBUTE VALUES BY DIFFERENT LLMS

We compare the impact of obtaining advantageous value ranges using GPT-4o and DeepSeek-R1 on optimizing the rationality-based reward, as shown in Table 5. Compare to the baseline model R1-

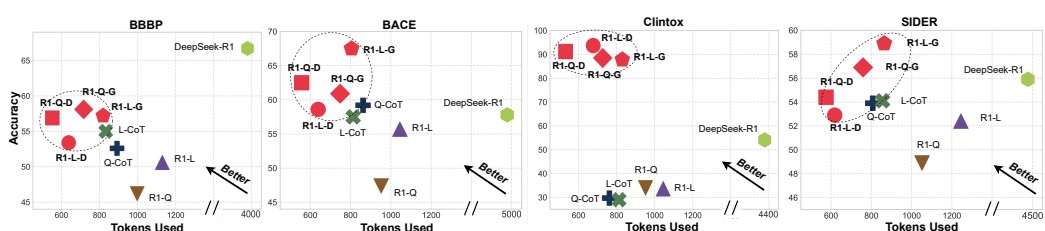

Figure 5: Accuracy and tokens used after tokenization for baselines and ours (in red). Q: Qwen2.5. L: LLaMA3.1. G: GRPO. D: DAPO. CoT: with chain-of-thought prompting.

| Models | Types | BBBP | BACE | ClinTox | AVG |
|---|---|---|---|---|---|
| GCN (Kipf & Welling, 2017) | Deep Neural Network | 0.6780 | 0.6893 | 0.8390 | 0.7354 |
| GIN (Xu et al., 2019) | Deep Neural Network | 0.6971 | 0.7346 | **0.8420** | 0.7579 |
| GraphMVP (Liu et al., 2022) | Deep Neural Network | 0.6780 | 0.7430 | 0.7900 | 0.7370 |
| AutoMolCo (Zhang et al., 2025) | LLM (GPT-3.5 Turbo) + Decision Tree | 0.6528 | 0.7074 | - | - |
| LLM4SD (Zheng et al., 2025) | LLM (Falcon-7B) + Decision Tree | 0.7135 | 0.7618 | 0.5000 | 0.6584 |
| R1-Distilled-Qwen2.5 (Guo et al., 2025) | LLM + Decision Tree | 0.6794 | 0.7799 | 0.7232 | 0.7275 |
| R1-Distilled-Qwen2.5 (Ours, GRPO) | LLM + Decision Tree | **0.7183*** | **0.7982*** | 0.8330* | **0.7832** |
| R1-Distilled-Qwen2.5 (Ours, DAPO) | LLM + Decision Tree | 0.7100* | 0.7842* | 0.8290* | 0.7744 |

Table 6: The AUC-ROC results by neural network and decision tree. AutoMolCo does report result on ClinTox. *: Statistical significance compared to R1-Distilled-Qwen2.5 with $p < 0.001$.

Distilled-Qwen2.5, both models led to significant performance improvements. The attribute values generated by GPT-4o slightly better results but the overall performance gap was marginal. The results show that both models can be powerful as a guidance for offering attribute-level information and help optimizing rationality-based reward.

### 5.4 COMPARISON OF THE LENGTH OF RESPONSE

In Figure 5, we compare the number of tokens used and the accuracy achieved by different models. Our models consistently uses significantly fewer tokens during inference while achieving high accuracy across all four tasks compared to baseline models, showing that our method makes the process of reasoning more efficiently and effectively.

### 5.5 ANALYSIS OF ATTRIBUTES USING DECISION TREE

To further validate the interpretability and effectiveness beyond textual output and accuracy, inspired by AutoMolCo (Zhang et al., 2025) and LLM4SD (Zheng et al., 2025), we use the top ten extracted attributes from training samples for each task and their values computed by RDKit as input features for training interpretable decision tree models (e.g., random forest), comparing the classification results using AUC-ROC metric. The pipeline for training decision tree and detailed settings are shown in Appendix C. The results are shown in Table 6. Our method achieves the best results on the BBBP and BACE, and also outperforms other decision tree baselines on the ClinTox and close to the uninterpretable neural classification methods based on GCN and GIN. These results demonstrate the relevance of the generated attributes during reasoning, which may help promote future developments in explainable molecular property prediction and related fields.

## 6 CONCLUSION

We propose AttriLens-Mol, an attribute-guided reinforcement learning framework for molecular property prediction. AttriLens-Mol integrates task-agnostic rule-based rewards with attribute-oriented rewards, enabling LLMs to optimize attribute selection and property reasoning. AttriLens-Mol achieves superior or comparable performance across in-distribution and out-of-distribution molecular property prediction datasets, comparing with both task-specific models or advanced LLMs via either pre-training, chain-of-thought prompting, or supervised fine-tuning.

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

## A  STATISTICS OF DATASET

The statistics of datasets are shown in Table 7.

| Datasets | #Train | #Valid | #Test | Type |
|---|---|---|---|---|
| BACE | 1,210 | 151 | 152 | Binary Classification |
| BBBP | 1,631 | 204 | 204 | Binary Classification |
| ClinTox | 1,182 | 148 | 148 | Binary Classification |
| SIDER | †1,141 | †143 | 143 | Binary Classification |
| ESOL | †902 | †113 | 113 | Regression |
| FreeSolv | †513 | †64 | 65 | Regression |

Table 7: The scaffold splits of the datasets used from MoleculeNet. †: Not used for OOD evaluation.

## B  IMPLEMENTATION DETAILS

TRL (von Werra et al., 2020) framework is used for training both R1-Distilled-Qwen2.5 and R1-Distilled-LLaMA3.1, with batch size 8 and steps 1000. During GRPO and DAPO, 8 candidate responses are generated for each input query with temperature of 0.6. Four RTX L40 GPUs with 48 GB memory are used, and the total training time is approximately five hours. The hyperparameters employed in our reinforcement learning experiments are summarized in Table 8.

| Parameter | Value | Parameter | Value |
|---|---|---|---|
| Learning Rate | 5e-5 | Warmup Ratio | 0 |
| Adam Beta1 | 0.9 | Logging Steps | 1 |
| Adam Beta2 | 0.99 | BF16 | True |
| Weight Decay | 0.1 | Gradient Accumulation | 4 |
| Max Grad Norm | 0.5 | Training Epochs | 1 |
| vLLM GPU Memory | 0.2 | Number of Generations | 8 |
| Max Prompt Length | 400 | Max Completion Length | 2000 |

Table 8: Hyperparameter configuration for reinforcement learning training.

## C  PIPELINE OF TRAINING DECISION TREE

The pipeline of training decision tree for attribute analysis is shown in Figure 6.s

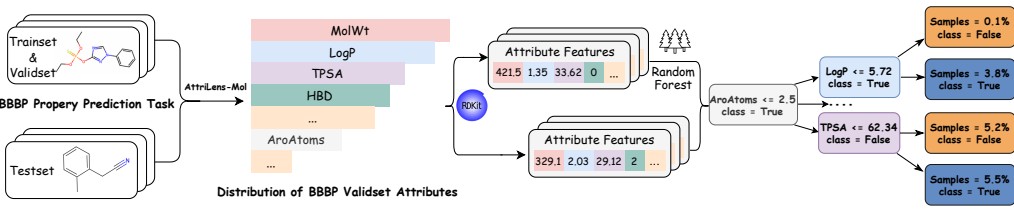

Figure 6: The framework of using features of attribute values for training a decision tree model.

## D  PROMPT TEMPLATE FOR MOLECULAR PROPERTY PREDICTION

To systematically guide the LLMs in predicting molecular properties, we designed a structured prompt template. This template aims to break down complex scientific reasoning tasks into manageable steps and compels the model to output its reasoning process and conclusions in a standardized format. The final structure of our prompt is illustrated in Figure 7.

---

**System Prompt:** You are an expert specializing in predicting {Task_Target}. Your task is to analyze whether the given molecule activates the {Task_Target}.
Please follow these steps:
1. Identify the most relevant molecular properties that influence {Task_Target} binding and activation.
2. Estimate the likely value of each property for the given molecule.
3. Explain whether each property promotes or inhibits {Task_Target} activation.
4. Combine these insights to give a final True/False judgment.
You must write your response using the following strict XML format: <think> Step-by-step reasoning with consideration on relevant attributes can be calculated using RDKit. For each attribute, provide its estimated value, and explain whether it promotes (improve) or inhibits (not improve) the target property. </think>, <name>List the attributes you used, each followed by : "promotes" or : "inhibits", separated by commas. For example: attribute A: promotes, attribute B: promotes, attribute C: inhibits. </name>, <answer>The final answer (e.g., true/false or specific values) based on your overall reasoning. </answer> .
Additional requirements:
- Your reasoning must be detailed but concise.
- True indicates that the molecule ACTIVATES the {Task_Target}.
- False indicates that the molecule does NOT activate the {Task_Target}.
**User Input:** The molecule you need to consider is {SMILES}.

---

Figure 7: Example of prompts for binary classification tasks.

# E  CASE STUDY

To qualitatively demonstrate the superiority of our model, we present comparative case studies in three distinct molecular property prediction tasks, including BBBP, BACE and Toxicity. As illustrated in Figures 8∼10, our model consistently delivers accurate results and interpretable, logically sound rationales, as reflected by its superior rationality reward $\mathcal{R}^{\text{rational}}$, while baseline models (e.g., Qwen2.5 and DeepSeek-R1) often produce incorrect or unreliably correct predictions due to flawed reasoning.

**BBBP (Figure 8): Superior Reasoning for Correct Prediction.**  In this task, only our model achieves the correct prediction, directly attributable to its perfect reasoning process ($\mathcal{R}^{\text{rational}}$ is 1.0). In contrast, the baseline models falter due to critical reasoning failures: DeepSeek-R1 and R1-Distilled-Qwen2.5 make incorrect assessments of how attributes affect BBBP ($\mathcal{R}^{\text{rational}}$ are 0.5 and 0.25, respectively), while Qwen2.5 enumerates attributes but fails to synthesize a conclusive effect. The latter two models also exhibit significant uncertainty in estimating basic properties, further undermining their reliability.

**BACE (Figure 9): Beyond Superficial Correctness with Causal Grounding.** Here, both Qwen2.5 and our model arrive at the correct answer. However, our model's perfect $\mathcal{R}^{\text{rational}}$ score underscores a fundamental difference: its reasoning provides clear, causally-grounded attribute analysis. Qwen2.5 offers only generic commentary without quantitative estimates or effect analysis ($\mathcal{R}^{\text{rational}}$ is 0), making its correct prediction appear fortuitous. R1-Distilled-Qwen2.5 fails structurally and quantitatively, while ours outperforms the well-formatted DeepSeek-R1 in reasoning coherence ($\mathcal{R}^{\text{rational}}$ are 1.0 and 0.66, respectively).

**Toxicity (Figure 10): Nuanced Analysis Overcoming Common Pitfalls.** This case highlights our model's ability to integrate nuanced attribute analysis for a correct prediction where other models fail. Although DeepSeek-R1 identifies relevant attributes, our model's deeper understanding, such as recognizing that the absence of sulfur atoms negates their contribution to toxicity, leads to a divergent and correct conclusion ($\mathcal{R}^{\text{rational}}$ is 1.0). Qwen2.5 and R1-Distilled-Qwen2.5 again suffer from formatting and incomplete reasoning, rendering their processes uninterpretable.

In summary, these case studies validate that our model's performance gain stems from its robust and interpretable reasoning framework. It consistently avoids the pitfalls of partial, uncertain, or causally-ungrounded reasoning seen in strong baselines, providing a credible explanation for its superior predictive accuracy.

# F  ETHICS STATEMENT

AI assistance was used solely to refine the writing of this paper. The model did not contribute to the generation of ideas, methodological design, or experimental work.

**BBBP Prediction Task**
Prompt: Your task is to predict the property of the given molecule. You must write your response using the following strict XML format: <think> Step-by-step reasoning with consideration on relevant attributes can be calculated using RDKit. For each attribute, provide its estimated value, and explain whether it promotes (improve) or inhibits (not improve) the target property. </think>, <name>List the attributes you used, each followed by : "promotes" or : "inhibits", separated by commas. For example: attribute A: promotes, attribute B: promotes, attribute C: inhibits. </name>, <answer>The final answer (e.g., true/false or specific values) based on your overall reasoning. </answer> The task is BBBP prediction, the molecule is CN(C(=O)Cc1ccc(Cl)c(Cl)c1)C1CCCC[C@H]1N1CCCC1.

**Response by Qwen2.5**
Assistant: <think> Molecular Weight: The molecular weight is around 367 g/mol. logP: While we don't have exact logP values, molecules with a logP < 5 are more likely to cross the BBB. The presence of Cl and amide groups may affect this. HB: The molecule has an amide group that can form hydrogen bonds, which might be favorable for BBB penetration. ... </think>
<name> Molecular Weight, HB, LogP, TPSA </name>
<answer> False </answer>
<label> True </label>
Reward: $\mathcal{R}^{format} = 1$, $\mathcal{R}^{correct} = 0$, $\mathcal{R}^{count} = 0$, $\mathcal{R}^{rational} = 0$

**Response by R1-Distilled-Qwen2.5**
Assistant: <think> Alright, let's tackle this problem step by step. ... 1. Lipophilicity (LogP) ... I'll need to calculate this using RDKit. 2. HBA: Molecules with strong hydrogen bonding tend to be more polar and less likely to cross the BBB. ... The amide group and the cyclic structure suggest that it might not cross the BBB efficiently. </think>
<name> LogP: inhibits, MolecularWeight: inhibits, HBA: inhibits, TPSA: inhibits </name>
<answer> False </answer>
<label> True </label>
Reward: $\mathcal{R}^{format} = 1$, $\mathcal{R}^{correct} = 0$, $\mathcal{R}^{count} = 0$, $\mathcal{R}^{rational} = 0.25$

**Response by DeepSeek-R1**
Assistant: <think> We are predicting BBBP ... LogP (3) in the range (1-4) -> promotes ... Molecular Weight (estimated 400 g/mol) -> below 500 -> promotes, ... , HBD (3) -> more than 2 reduce permeability, ... PSA (70A²) -> high PSA (> 60A²) -> inhibit .... overall i think inhibit factors overrides other factors, thus the result might be False ... </think>
<name> LogP: promotes, MolecularWeight: promotes, HBD: inhibits, PSA: inhibits </name>
<answer> False </answer>
<label> True </label>
Reward: $\mathcal{R}^{format} = 1$, $\mathcal{R}^{correct} = 0$, $\mathcal{R}^{count} = 0$, $\mathcal{R}^{rational} = 0.5$

**Response by Ours**
Assistant: <think> Alright, let's tackle this problem step by step ... From what I remember, the main factors are Mol Weight, LogP and TPSA that might interfere with BBB permeability. And now i will estimate the attributes values .... Mol Weight (350 Da) ... LogP (5) ... TPSA (30A²) ... The Molecular Weight is below 500, which is favorable for BBB permeability, A LogP around 5 is borderline; higher LogP generally increases lipophilicity but too high may reduce solubility. In this case, it slightly inhibits BBB permeability. The TPSA less than 90 $\text{Å}^2$ is favorable for BBB penetration. Therefore, TPSA here promotes BBB permeability. Therefore the answer would be True. </think>
<name> LogP: inhibits, TPSA: promotes, MolecularWeight: promotes </name>
<answer> True </answer>
<label> True </label>
Reward: $\mathcal{R}^{format} = 1$, $\mathcal{R}^{correct} = 2$, $\mathcal{R}^{count} = 0$, $\mathcal{R}^{rational} = 1$

Figure 8: Case study examples comparing the responses of Qwen2.5, R1-Distilled-Qwen2.5, DeepSeek-R1, and ours on BBBP property prediction task.

**BACE Prediction Task**
Prompt: Your task is to predict the property of the given molecule. You must write your response using the following strict XML format: <think>Step-by-step reasoning with consideration on relevant attributes can be calculated using RDKit. For each attribute, provide its estimated value, and explain whether it promotes (improve) or inhibits (not improve) the target property. </think>, <name>List the attributes you used, each followed by : "promotes" or : "inhibits", separated by commas. For example: attribute A: promotes, attribute B: promotes, attribute C: inhibits. </name>, <answer>The final answer (e.g., true/false or specific values) based on your overall reasoning. </answer> The task is BACE prediction, the molecule is Nc1nc2ccccc2cc1CCC(=O)NCC1CCCCC1.

**Response by Qwen2.5**
Assistant: <think> To predict whether this molecule has inhibitory activity against BACE-1 ... 1. Hydrophobicity: The presence of multiple aromatic rings and alkyl chains suggests good hydrophobic interactions, which are important for binding to BACE-1. 2. Functional Groups: Acylamine (CONH) This group can form hydrogen bonds with the enzyme's active site ... Final Answer: True. </think>
<name> Hydrophobicity, Func Groups </name>
<answer> True </answer>
<label> True </label>
Reward: $\mathcal{R}^{\text{format}} = 1$, $\mathcal{R}^{\text{correct}} = 2$, $\mathcal{R}^{\text{count}} = -1$, $\mathcal{R}^{\text{rational}} = 0$

**Response by R1-Distilled-Qwen2.5**
Assistant: <think> Alright, let me try to figure this out. ... 1. Hydrogen Bond Donors/Acceptors ... However, without specific data, it's hard to quantify ... 2. Aromatic Ring Substituents: which might shield the aromatic ring, making it less likely to interact with the BACE enzyme's aromatic region. ... 3. Amide Groups: which might affect the binding affinity of the molecule to BACE. ... 4. Ethylenic Groups: ... It's a bit ambiguous without more structural details. </think>
<label> True </label>
Reward: $\mathcal{R}^{\text{format}} = -2$, $\mathcal{R}^{\text{correct}} = 0$, $\mathcal{R}^{\text{count}} = -1$, $\mathcal{R}^{\text{rational}} = 0$

**Response by DeepSeek-R1**
Assistant: <think> Step-by-step reasoning: HBD: 2, BACE-1 inhibitors typically require > 3 HBDs ... This inhibits prediction. 2. HBA: Estimated 3. Sufficient but not optimal. Neutral effect. 3. MolecularWeight: :324. Within druglike range 500 ... </think>
<name> HBD: inhibits, HBA: promotes, Molecular Weight: promotes, LogP: promotes, AmideGroup: inhibits, AromaticRings: inhibits </name>
<answer> False </answer>
<label> True </label>
Reward: $\mathcal{R}^{\text{format}} = 1$, $\mathcal{R}^{\text{correct}} = 0$, $\mathcal{R}^{\text{count}} = 0$, $\mathcal{R}^{\text{rational}} = 0.66$

**Response by Ours**
Assistant: <think> Okey, so I need to figure out whether the given molecule inhibits BACE-1 ... 1. LogP: (3.5) Moderate LogP is likely good, ... so this would improve inhibition. 2. TPSA: (20-25A²) A moderate TPSA is better, so this might not improve inhibition if it's too high. ... I think the molecule would inhibit BACE. </think>
<name> LogP: promotes, TPSA: inhibits, Molecular Weight: promotes, H-Bonding: promotes, Ringnumber: promotes </name>
<answer> True </answer>
<label> True </label>
Reward: $\mathcal{R}^{\text{format}} = 1$, $\mathcal{R}^{\text{correct}} = 2$, $\mathcal{R}^{\text{count}} = 0$, $\mathcal{R}^{\text{rational}} = 1$

Figure 9: Case study examples comparing the responses of Qwen2.5, R1-Distilled-Qwen2.5, DeepSeek-R1, and ours on BACE property prediction task.

**Toxic Prediction Task**
Prompt: Your task is to predict the property of the given molecule. You must write your response using the following strict XML format: <think>Step-by-step reasoning with consideration on relevant attributes can be calculated using RDKit. For each attribute, provide its estimated value, and explain whether it promotes (improve) or inhibits (not improve) the target property. </think>, <name>List the attributes you used, each followed by : "promotes" or : "inhibits", separated by commas. For example: attribute A: promotes, attribute B: promotes, attribute C: inhibits. </name>, <answer>The final answer (e.g., true/false or specific values) based on your overall reasoning. </answer> The task is Toxic prediction, the molecule is C[NH+]1C[C@H](C(=O)N[C@]2(C)O[C@@]3(O)[C@@H]4CCCN4C(=O)[C@H].

**Response by Qwen2.5**
Assistant: <think> Step-by-step reasoning ... 1. TPSA: 140A (which is moderate but still within a range) -> inhibits, Molecular Weight: 562g/mol, higher molecular weights can sometimes reduce penetration and bioavailability but do not directly correlate with increased toxicity in all cases ... 3. Heavy Atom Count: the molecule has 40 heavy atoms possibly leading to toxic effects ... TPSA: inhibits, Molecular Weight: inhibits, HAC: promotes, Ring Count promotes. </think>
Answer = True
<label> False </label>
Reward: $\mathcal{R}^{format}$ = -2, $\mathcal{R}^{correct}$ = 0, $\mathcal{R}^{count}$ = -1, $\mathcal{R}^{rational}$ = 0

**Response by R1-Distilled-Qwen2.5**
Assistant: <think> Alright, so I need to figure out whether this molecule is toxic or not ... 1. Functional Groups: the molecule contains amide bonds (N-C=O), hydroxyl groups (-OH), aniline-like amino groups (-NH2), and some aromatic rings with methyl substituents. 2. HBD: Influencing solubility but potentially leading to increased bioavailability... So, after weighing these considerations, my judgment is that this molecule is not highly toxic. </think>
Answer = True
<label> False </label>
Reward: $\mathcal{R}^{format}$ = -2, $\mathcal{R}^{correct}$ = 0, $\mathcal{R}^{count}$ = -1, $\mathcal{R}^{rational}$ = 0

**Response by DeepSeek-R1**
Assistant: <think> We are to predict if the molecule is toxic ... 1. Molecular Weight: 600. High molecular weight can hinder excretion, leading to accumulation and potential toxicity. -> promotes. 2. LogP: 2.5. Moderate lipophilicity ... increasing toxicity risk ... </think>
<name> LogP: promotes, MolecularWeight: promotes, HCount: promotes, TPSA: promotes, NHD: inhibits </name>
<answer> True </answer>
<label> False </label>
Reward: $\mathcal{R}^{format}$ = 1, $\mathcal{R}^{correct}$ = 0, $\mathcal{R}^{count}$ = 0, $\mathcal{R}^{rational}$ = 0.6

**Response by Ours**
Assistant: <think> ... 1. Molecular Weight: 600-700 Heavier molecules can sometimes be more toxic, but this also depends on the functional groups and structure ... 2. S Count: No sulfur atoms mean this property doesn't contribute to toxicity in this case ... the molecule lacks explicit high-toxicity warning structures (False) ... </think>
<name> MolecularWeight: promotes, S count: inhibits, TPSA: promotes </name>
<answer> False </answer>
<label> False </label>
Reward: $\mathcal{R}^{format}$ = 1, $\mathcal{R}^{correct}$ = 2, $\mathcal{R}^{count}$ = 0, $\mathcal{R}^{rational}$ = 1

Figure 10: Case study examples comparing the responses of Qwen2.5, R1-Distilled-Qwen2.5, DeepSeek-R1, and ours on toxic property prediction task.

