# OpenReview forum: "AttriLens-Mol: Attribute Guided Reinforcement Learning for  Molecular Property Prediction with Large Language Models"
_ICLR.cc/2026/Conference — Submitted to ICLR 2026_

### Official Review · Reviewer_anam · 2025-10-28

**Soundness:** 3
**Presentation:** 3
**Contribution:** 3
**Rating:** 4
**Confidence:** 3

**Summary:**

Summary:
This paper introduces AttriLens-Mol, an attribute-guided reinforcement learning (RL) framework that enhances molecular property prediction in Large Language Models. It uses three specific rewards to guide LLM’s reasoning incluyding a format reward, a count reward, and a rationality reward. The model significantly boosts the performance of 7B models, allowing them to achieve results comparable or superior to much larger models.

**Strengths:**

Pros:
- The authors has introduced a new model called AttriLens-Mol, to enhance molecular property prediction with LLMs
- The model has achieved better performance compared with 7B models baselines, and achieve comparable results to much large models

**Weaknesses:**

Cons:
- The rationality reward is not self-contained. It relies on judgment from advanced LLMs like GPT-4O and DeepSeek-R1. This way may creates a dependency, meaning the performance is tied to the accuracy and knowledge of external models.
- More benchmark datasets are encouraged to added and tested. Currently, the model is only tested on six datasets in MoleculeNet. Why other benchmark datasets is not included, e.g., HIV, and Tox21.
- Compared to standard supervised fine-tuning, the reinforcement learning approach is computationally expensive. For each training sample, the model must generate a group of 8 candidate responses. Each of these 8 responses must then be evaluated by the four separate reward functions

**Questions:**

Shown in Weaknesses

---

> ### Author Response · Authors · 2025-11-27
> **Response**
>
> We sincerely thank the reviewer for their positive assessment of our work. We particularly appreciate the recognition of the soundness, presentation, and contribution all being rated as "good". We cherish the chance to resolve your concerns in details.
>
> > **W1**. The rationality reward is not self-contained. It relies on judgment from advanced LLMs like GPT-4O and DeepSeek-R1. This way may creates a dependency, meaning the performance is tied to the accuracy and knowledge of external models.
>
> To our knowledge, there is currently no definitive consensus in the scientific community regarding the precise correlation between molecular attributes and biological activity. The design of our rationality reward mechanism is conceptually aligned with approaches such as LLM4SD [1] and AutoMolCo [2], which similarly utilize advanced LLMs like GPT-4 for knowledge extraction in molecular property prediction tasks. Thereby, we extract knowledge from external LLMs rather than specific computational results, leveraging the abstract understanding and generalization capabilities of LLMs to construct a relatively reliable bioinformatic knowledge source.
>
>
> > **W2**. More benchmark datasets are encouraged to added and tested. Currently, the model is only tested on six datasets in MoleculeNet. Why other benchmark datasets is not included, e.g., HIV, and Tox21.
>
> We follow LLaSMol [1] and AutoMolCo [2] for dataset selection. From the additional results below, with comparison to R1-Distilled-Qwen2.5-7B model, our approach achieves a remarkable increase of +18.48 points on the HIV dataset and +15.20 points on the Tox21 dataset. Moreover, it performs on par with the GPT-4o (with **>100B** parameters) on the two tasks. These results confirm that our attribute-guided reinforcement learning framework can effectively generalize to diverse and challenging molecular property prediction tasks beyond our initial training distribution. We will include these comprehensive results in the final manuscript to provide a more complete evaluation of our method's capabilities.
>
> We appreciate this suggestion, as these additional experiments have further strengthened our contribution by demonstrating the robustness and generalization power of our method across diverse benchmark conditions. We would like to provide more comprehensive comparison results with more time in the future.
>
>
> | Model                                | HIV       | Tox21 |
> |--------------------------------------|-----------|--------------|
> | GPT-4o                               | 59.60 | 50.19        |
> | Qwen2.5-7B                           | 42.77     | 36.02        |
> | R1-Distilled-Qwen2.5-7B              | 41.21     | 33.59        |
> | R1-Distilled-Qwen2.5-7B (Ours, GRPO) | 59.69     | 48.79        |
>
> > **W3**. Compared to standard supervised fine-tuning, the reinforcement learning approach is computationally expensive. For each training sample, the model must generate a group of 8 candidate responses. Each of these 8 responses must then be evaluated by the four separate reward functions
>
> Our proposed RL approach indeed requires generating multiple candidate responses per sample, while we have implemented several key design choices that ensure training efficiency:
>
> First, unlike SFT which primarily learns to imitate existing patterns, our GRPO-based method actively optimizes for multiple complex objectives simultaneously, including chemical rationality, attribute relevance, and reasoning coherence. This represents a qualitatively different learning paradigm that justifies the additional per-step computation.
>
> Second, all four reward functions are designed for efficiency, utilizing fast RDKit descriptor calculations and rule-based checks that support parallel processing. Our method achieves convergence with just ~4,000 multi-task samples in a single training epoch, compared to SFT which often requires multiple epochs and larger task-specific datasets. The 8-candidate sampling provides sufficient diversity for effective policy optimization while remaining computationally manageable.
>
> Third, our complete training process completes in approximately 5 hours using 4×RTX L40 GPUs, demonstrating practical feasibility. The significant performance gains across both in-distribution and challenging out-of-distribution tasks (as shown in Tables 2, 3, and our added HIV/Tox21 results) confirm that this computational investment yields substantial returns in model capability. We believe this represents a favorable trade-off, as the additional computation enables genuine reasonin
>
>
> Reference:
> [1] Zheng, Yizhen, et al. Large language models for scientific discovery in molecular property prediction. Nature Machine Intelligence 2025.
> [2] Zhang, Zimin, et al. Automated molecular concept generation and labeling with large language models. COLING 2025.
>
> ---
>
> Thank you again for your time and effort. We remain open to any further suggestions or requests you may have.

---

### Official Review · Reviewer_BsNe · 2025-10-30

**Soundness:** 3
**Presentation:** 3
**Contribution:** 2
**Rating:** 6
**Confidence:** 4

**Summary:**

This paper presents AttriLens-Mol, an attribute-guided reinforcement learning framework designed to enhance molecular property prediction using LLMs. AttriLens-Mol introduces three complementary reward functions—format, count, and rationality—to systematically align LLM reasoning with molecular attributes in a chemically interpretable manner. Extensive experiments on both in-distribution and out-of-distribution MoleculeNet benchmarks (BBBP, BACE, ClinTox, etc.) demonstrate that the proposed approach consistently outperforms supervised and prompting-based baselines. Additionally, decision-tree analyses reveal that the attributes generated by the model can serve as interpretable features for property classification.

**Strengths:**

The paper presents an innovative idea by using attribute-level rewards to align LLM reasoning with molecular properties, which is both novel and meaningful. This is supported by a solid methodology, with the design of three coherent and clearly explained rewards. Furthermore, the comprehensive evaluation compares multiple models (Qwen, LLaMA) and algorithms (GRPO, DAPO), demonstrating strong and consistent gains. Interpretability is reinforced through decision-tree analysis based on extracted attributes, effectively supporting the claim of explainable reasoning. Overall, the paper is well-written, clear, structured, and easy to follow.

**Weaknesses:**

- The reliance on closed-source LLMs (GPT-4o and DeepSeek-R1) for generating advantageous attribute ranges raises reproducibility concerns; clarifying the derivation process and assessing whether open-source models yield comparable results would improve the paper.

- The limited experimental scale—approximately 4,000 samples—constrains the empirical scope; including a brief discussion or small-scale analysis of performance on larger datasets would strengthen the credibility and generalization claims.

**Questions:**

1. Why do the authors need to extract relevant attribute information from GPT-4o or DeepSeek-R1? Is there a fixed standard for defining these attribute ranges, rather than relying on LLM-generated values? Could this approach introduce concerns regarding reproducibility?

2. Why did the authors choose to use approximately 4,000 training samples from the BBBP, BACE, and ClinTox datasets? Would increasing or decreasing the amount of training data have a significant impact on the results?

3. Are these advantageous ranges defined separately for each molecular property, or are they shared globally across all tasks? If a single unified set of ranges is used, could this lead to potential attribute bias or information leakage? Additionally, are the model’s results sensitive to small perturbations in these ranges?

4. Since the reward is computed using discrete matching (as shown in Equation 4), have the authors considered the potential issue of reward sparsity that this formulation might introduce?

5. Providing clearer descriptions of the reinforcement learning hyperparameters and settings — such as the KL coefficient, clipping parameter, optimizer and learning rate schedule, reference policy, context length, decoding strategy, seed control, and safeguards against degenerate advantage normalization — would greatly improve the reproducibility and robustness of the work.

---

> ### Author Response · Authors · 2025-11-27
> **Response (Part 1)**
>
> Thank you so much for your valuable comments. Below are our responses.
>
> > **W1&Q1**. The reliance on closed-source LLMs (GPT-4o and DeepSeek-R1) for generating advantageous attribute ranges raises reproducibility concerns; clarifying the derivation process and assessing whether open-source models yield comparable results would improve the paper.
>
> To our knowledge, there is currently no definitive consensus in the scientific community regarding the precise correlation between molecular attributes and biological activity. The design of our method is inspired by pioneering works such as LLM4SD [1] and AutoMolCo [2], which explore how to extract prior knowledge from advanced LLMs and utilize it as computable features or guidance signals to enhance model performance.
>
> In line with this research direction, our core idea is to transform the chemical commonsense knowledge embedded in LLMs, which is difficult to control directly, into a structured and verifiable reward signal, so as to guide smaller models toward more reliable reasoning.
>
> Furthermore, we conducted comparative experiments between DeepSeek-R1 and ChatGPT. As shown in Table 5, the results demonstrate the strong robustness of our approach.
>
> > **W2**. The limited experimental scale—approximately 4,000 samples—constrains the empirical scope; including a brief discussion or small-scale analysis of performance on larger datasets would strengthen the credibility and generalization claims.
>
> Our experimental design is inspired by the successful application of RL in other domains, which demonstrates that moderate-scale RL training on high-quality data can lead to strong generalization capabilities. Take the mathematical reasoning task as an example, DeepSeek-Math [3] and Goal-conditioned RM [4] have achieved breakthrough performance by applying RL to datasets with only thousands of samples (e.g., GSM8K [5]).
>
> These precedents indicate that the core advantage of RL lies in efficiently eliciting the model's inherent knowledge through reward signals, rather than relying on large-scale data memorization. Based on the characteristics of our task, we perfer several thousand high-quality data samples for training. Experimental results show that all four of our reward function curves quickly converged during the early stages of training (within approximately 1,000 steps), indicating that the policy of model stabilized rapidly. This further validates the rationality and efficiency of our data scale selection.
>
> > **Q2**. Why did the authors choose to use approximately 4,000 training samples from the BBBP, BACE, and ClinTox datasets? Would increasing or decreasing the amount of training data have a significant impact on the results?
>
> We follow the dataset selection of AutoMolCo [2] and LlasMol [6] on MoleculeNet. As mentioned in Q2, we specifically chose BBBP, BACE, and Clintox for our experiments to ensure that the dataset sizes remained within a reasonable range.
>
> > **Q3**. Are these advantageous ranges defined separately for each molecular property, or are they shared globally across all tasks? If a single unified set of ranges is used, could this lead to potential attribute bias or information leakage? Additionally, are the model’s results sensitive to small perturbations in these ranges?
>
> We obtain distinct advantageous ranges from advanced LLMs for different tasks, which enables the model to effectively differentiate between tasks and to develop a more nuanced understanding of their unique characteristics. In practice, the obtained advantageous ranges from LLMs is relatively stable, thus the impact of small perturbations is minimal.

---

> > ### Author Response · Authors · 2025-11-27
> > **Response (Part 2)**
> >
> > > **Q4**. Since the reward is computed using discrete matching (as shown in Equation 4), have the authors considered the potential issue of reward sparsity that this formulation might introduce?
> >
> > Our design intentionally leverages group-wise comparison in GRPO training. By evaluating multiple responses per prompt, the discrete matching creates relative differences that provide sufficiently dense gradients for optimization. The stable convergence observed in experiments confirms that this formulation delivers effective learning signals in practice.
> >
> > > **Q5**. Providing clearer descriptions of the reinforcement learning hyperparameters and settings — such as the KL coefficient, clipping parameter, optimizer and learning rate schedule, reference policy, context length, decoding strategy, seed control, and safeguards against degenerate advantage normalization — would greatly improve the reproducibility and robustness of the work.
> >
> > The key hyperparameters used in our experiments is shown below (see also the appendix B of the revised version). To ensure reproducibility, we commit to fully open-sourcing our project upon acceptance of the paper, which will include complete implementation details and experimental configurations.
> >
> > | Parameter                   | Value | Parameter                   | Value |
> > |-----------------------------|-------|-----------------------------|-------|
> > | Learning Rate               | 5e-5  | Warmup Ratio                | 0     |
> > | Adam Beta1                  | 0.9   | Logging Steps               | 1     |
> > | Adam Beta2                  | 0.99  | BF16                        | True  |
> > | Weight Decay                | 0.1   | Gradient Accumulation       | 4     |
> > | Max Grad Norm               | 0.5   | Training Epochs             | 1     |
> > | vLLM GPU Memory             | 0.2   | Number of Generations       | 8     |
> > | Max Prompt Length           | 400   | Max Completion Length       | 2000  |
> >
> > Reference:
> > [1] Zheng, Yizhen, et al. Large language models for scientific discovery in molecular property prediction. Nature Machine Intelligence 2025.
> > [2] Zhang, Zimin, et al. Automated molecular concept generation and labeling with large language models. COLING 2025.
> > [3] Shao, Zhihong, et al. DeepSeekMath: Pushing the limits of mathematical reasoning in open language models. arXiv 2402.03300.
> > [4] Nath, Vaskar, et al. Learning goal-conditioned representations for language reward models. NeurIPS 2024. 117070-117108.
> > [5] Cobbe, Karl, et al. Training verifiers to solve math word problems. arXiv 2110.14168.
> > [6] Yu, Botao, et al. LlaSMol: Advancing large language models for chemistry with a large-scale, comprehensive, high-quality instruction tuning dataset. COLM 2024.
> >
> > ---
> >
> > Thank you again for your time and effort. We remain open to any further suggestions or requests you may have.

---

### Official Review · Reviewer_i7Ns · 2025-10-31

**Soundness:** 3
**Presentation:** 3
**Contribution:** 2
**Rating:** 6
**Confidence:** 4

**Summary:**

This paper proposes a reinforcement learning (RL) framework, AttriLens-Mol, to fine-tune large language models (LLMs) for molecular property prediction using "attribute-guided" rewards. The core contribution is the "rationality reward," which uses a stronger "guidance LLM" (like GPT-4O) to define "advantageous ranges" and then combines RDKit computations to "factcheck" the reasoning steps generated by the 7B model. The paper claims this method "elicits inherent knowledge" and achieves strong performance on in-distribution (ID) and out-of-distribution (OOD) tasks. However, the methodology is closer to knowledge distillation (from GPT-4O to the 7B model) rather than the "discovery" or "elicitation" of new knowledge. Furthermore, the framework is critically dependent on RDKit's fixed descriptors, limiting its generality and application in genuine scientific discovery.

**Strengths:**

The empirical results of AttriLens-Mol on OOD  regression tasks (Table 2) are indeed impressive. The performance on OOD tasks (ESOL/FreeSolv) is significantly better than the SFT baselines, which is a very strong empirical argument.

The decision tree analysis in Table 6 is a welcome addition, as it attempts (and succeeds) to quantitatively prove that the attributes (features) generated by the LLM are predictive, not just "plausible-sounding" text.

**Weaknesses:**

1. The core "rationality reward" is entirely dependent on an external "guidance LLM" (like GPT-4O) to define "advantageous ranges." This raises a methodological concern: the 7B model is not "eliciting inherent knowledge" as claimed, but rather performing high-efficiency knowledge distillation. The most critical "chemical common sense" is injected externally, anchoring the model's performance to the quality of the guidance LLM.

2. The "rationality reward" mechanism is hard-coded to RDKit's 53 descriptors, severely restricting its applicability. The framework will fail on tasks requiring features beyond RDKit's scope . Consequently, the paper's title "...for Molecular Property Prediction" is an overclaim, as it is limited to " RDKit-descriptor-predictable" properties and not genuine scientific discovery.

3. The "count reward" (rewarding [3, 10] attributes) is a heuristic, ad-hoc design lacking principled argumentation. This one-size-fits-all constraint is problematic: it could penalize simple tasks (requiring <3 attributes) or hinder complex ones (requiring >10). The ablation study may only show that the model learned to "fill the quota," not "autonomously select the most relevant attributes" for the task.

**Questions:**

1. Can you provide an experiment quantifying the
impact of the "guidance LLM" (e.g., GPT-4O) quality on the final performance? For example, how much
would performance drop if a "weaker" guidance LLM (like LLaMA3-8B) is used to generate the "
advantageous ranges"? This is critical for disentangling whether the model is "learning to reason" or "
distilling knowledge."

2. What is your view on the framework's scalability beyond the fixed RDKit
descriptors? If a task's key attributes are unknown or non-RDKit-computable (e.g., predicting quantum-
chemical properties), how would the framework adapt? Or do you acknowledge that the framework is not
applicable to such tasks?

3. The design of the count reward seems counter-intuitive as it limits the model's free exploration. Did you experiment with a "softer" constraint, such as a sparsity penalty (i.e., a small negative reward for each additional attribute), to replace the hard 3, 10 range? Could this allow the model to autonomously learn the optimal number of attributes, rather than relying on the manually set
range?

---

> ### Author Response · Authors · 2025-11-27
> **Response**
>
> We thank the reviewer for the positive recognition of our contributions.
>
> > **Q1**. Can you provide an experiment quantifying the impact of the "guidance LLM" (e.g., GPT-4O) quality on the final performance? For example, how much would performance drop if a "weaker" guidance LLM (like LLaMA3-8B) is used to generate the " advantageous ranges"? This is critical for disentangling whether the model is "learning to reason" or " distilling knowledge".
>
> We indeed conducted the experiment you suggested to quantify the impact of the guidance quality of LLM. As shown in the table below, the significant performance drop of LLaMA3.1 stems from its generation of chemically unreasonable "advantageous ranges" in BBBP task. Also, GPT-4o and DeepSeek-R1 provided reasonable molecular weight intervals of (100,450) and (100,500) respectively, while LLaMA3.1 provided an illogical range of (400,500), which contradicts established chemical knowledge about blood-brain barrier penetration.
>
> | Model                                       | BBBP |
> |---------------------------------------------|------|
> | R1-Distilled-Qwen2.5-7B                     | 51.8 |
> | R1-Distilled-Qwen2.5-7B (**LLaMA3.1-8B**, GRPO) | **49.0** |
> | R1-Distilled-Qwen2.5-7B (DeepSeek-R1, GRPO) | 57.3 |
> | R1-Distilled-Qwen2.5-7B (GPT-4o, GRPO)      | 58.1 |
>
> This evidence indicates that different model selections can significantly impact experimental outcomes. Choosing advanced models enables better summarization and integration of existing knowledge, which aligns with the strategies of AutoMolCo [1] and LLM4SD [2]. Smaller models are more likely to generate unreasonable content that even conflicts with established common sense, leading to biased training results.
>
> Overall, we indeed distilled useful knowledge from advanced LLMs, however, the attribute value related to knowledge is not directly equal to the final label of property prediction tasks. Our method use the information of attribute value provided by LLMs and facilite the reasoning ability for molecular properties, which actually surpass the "teacher" of advanced LLMs.
>
> > **Q2**. What is your view on the framework's scalability beyond the fixed RDKit descriptors? If a task's key attributes are unknown or non-RDKit-computable, how would the framework adapt? Or do you acknowledge that the framework is not applicable to such tasks?
>
> RDKit is one of the widely-used open-source toolkit in bioinformatics, which has already supplies rich molecular features. Our method is also not restrict to specific tools and the core architecture of our framework is decoupled. Therefore, to apply our method to tasks beyond RDKit descriptors, we only need to replace the computation backend from RDKit with the corresponding dedicated tools (e.g., Gaussian [3], ORCA [4]), while the core functions and workflow of LLM remain unchanged.
>
> > **Q3**. The design of the count reward seems counter-intuitive as it limits the model's free exploration. Did you experiment with a "softer" constraint, such as a sparsity penalty (i.e., a small negative reward for each additional attribute), to replace the hard 3, 10 range?
>
> A primary objective in the initial phase of our research was to establish a stable and interpretable "rational reward" mechanism. Given the absence of a definitive consensus on the optimal number of relevant molecular attributes for each prediction task, we follow existing work such as AutoMolCo [1] and MPCD [5], which utilized 7-9 molecular attributes as features and invloved 12 different attribute, respectively (see line 200). Our designed range of 3-10 attributes builds upon this foundation, providing a broader exploration space while adhering to the chemical intuition that effective suggestions should be neither overly vague (e.g., generating only 1 attribute) nor excessively trivial (e.g., generating 20 attributes).
>
> We thoroughly considered softer constraints, such as sparsity penalties. However, the negative reward for each additional attribute will lead to insufficient attribute-based discussion. Thereby the "hard constraints" in proper ranges is still effective for general property prediction tasks.
>
> Reference:
> [1] Zhang, Zimin, et al. Automated molecular concept generation and labeling with large language models. COLING 2025.
> [2] Zheng, Yizhen, et al. Large language models for scientific discovery in molecular property prediction. Nature Machine Intelligence 2025.
> [3] Frisch, M. J. et al. Gaussian 16, Revision B.01. Gaussian, Inc., Wallingford CT, 2016.
> [4] Neese, Frank. The ORCA program system. Wiley Interdisciplinary Reviews: Computational Molecular Science 2.1 2012.
> [5] MPCD: A multitask graph transformer for molecular property prediction by integrating common and domain knowledge.  Journal of Medicinal Chemistry 2024.
>
> ---
> Thank you again for your time and effort. We remain open to any further suggestions or requests you may have.

---

### Official Review · Reviewer_p6WF · 2025-11-01

**Soundness:** 3
**Presentation:** 2
**Contribution:** 1
**Rating:** 4
**Confidence:** 4

**Summary:**

This paper proposes a reward design for reinforcement learning on molecular property prediction tasks using LLMs. Specifically, it defines two types of rewards: (1) task-agnostic rewards, which include a format reward and a correctness reward, and (2) attribute-oriented rewards consisting of a count reward that enforces the number of attributes to fall within a predefined range (e.g., 3–10), and a rationality reward that encourages selecting relevant molecular attributes and predicting them accurately. Using the proposed rewards, the model is fine-tuned separately with GPRO and DAPO. Experimental results show that this reward design improves predictive performance on both in-distribution and out-of-distribution tasks, while providing interpretable rationales.

**Strengths:**

* The reward function is well-designed to plant fine-grained reasoning capabilities and interpretability.
* The ablation study demonstrates the effectiveness of each reward component.

**Weaknesses:**

* The performance of the fine-tuned model heavily depends on the completeness and quality of the predicted attributes. In other words, if important but non-standard attributes are not included in the attribute list or the predicted values are incorrect, the LLM may fail to provide correct and complete answers. I suggest analyzing the number of correct attributes or the distribution of the selected (promoted) attributes for each task.
* Although the proposed reward design improves predictive performance, it still lags behind LLMs such as the GPT series and the DeepSeek series on some tasks. Moreover, there is no discussion or experimental comparison with the state-of-the-art model TxGemma [1], which is trained for molecular property tasks.
* The instructions used for LLMs are not precisely described, limiting the reproducibility of this work. In addition, using broader datasets such as ToxCast or HIV would further strengthen the contribution of the proposed reward.

[1] Wang et al., "TxGemma: Efficient and Agentic LLMs for Therapeutics", ArXiv, 2025.

**Questions:**

* For each task, could you provide the distributions of promoted and inhibited attributes during reasoning? How accurate are the predicted attributes?
* Are the individual reward components simply summed without weighting?

---

> ### Author Response · Authors · 2025-11-27
> **Response (Part 1)**
>
> Thank you for your valuable comments! We are glad to hear that you acknowledge the importance of well-designed reward function, our comprehensive comparison and ablation study. We cherish the chance to resolve your concerns in details.
>
> > **W1&Q1**. The performance of the fine-tuned model heavily depends on the completeness and quality of the predicted attributes. In other words, if important but non-standard attributes are not included in the attribute list or the predicted values are incorrect, the LLM may fail to provide correct and complete answers. I suggest analyzing the number of correct attributes or the distribution of the selected (promoted) attributes for each task.
>
> Following your suggestion, we have added a quantitative analysis of the attribute distribution of our GRPO-enhanced LLM across different tasks. As shown in the tables below, we have analyzed the relationship between the number of attributes selected by the model and the corresponding prediction accuracy for each task.
>
>
> ## BBBP Task:
>
> | Property Count | Attrilens-Mol Samples | Baseline Samples | Attrilens-Mol Acc. | Baseline Acc. | Attrilens-Mol Percentage | Baseline Percentage |
> |---------------|-------------------|--------------|----------------|--------------|----------------|--------------|
> | 1  | —  | 10 | —       | 50.00% | —     | 4.90% |
> | 2  | 1  | 35 | 100.00% | 51.43% | 0.49% | 17.16% |
> | 3  | 20 | 42 | 45.00%  | 52.38% | 9.80% | 20.59% |
> | 4  | **80** | 38 | **63.75%** | 50.00% | **39.22%** | 18.63% |
> | 5  | 74 | 32 | 55.41% | 50.00% | 36.27% | 15.69% |
> | 6  | 14 | 25 | 50.00% | 48.00% | 6.86%  | 12.25% |
> | 7  | 6  | 15 | 16.67% | 46.67% | 2.94%  | 7.35% |
> | 8  | 4  | 7  | 75.00% | 42.86% | 1.96%  | 3.43% |
> | 9  | 2  | —  | 50.00% | —      | 0.98%  | — |
> | 11 | 2  | —  | 50.00% | —      | 0.98%  | — |
> | 14 | 1  | —  | 100.00% | —     | 0.49%  | — |
>
>
> ## BACE Task:
>
> | Property Count | Attrilens-Mol Samples | Baseline Samples | Attrilens-Mol Acc. | Baseline Acc. | Attrilens-Mol Percentage | Baseline Percentage |
> |----------------|-------------------|------------------|----------------|--------------|----------------|--------------|
> | 1  | —  | 8  | —       | 37.50% | —      | 5.26% |
> | 2  | —  | 12 | —       | 58.33% | —      | 7.89% |
> | 3  | 23 | 18 | 47.83%  | 44.44% | 15.13% | 11.84% |
> | 4  | 21 | 16 | 57.14%  | 37.50% | 13.82% | 10.53% |
> | **5** | **71** | 22 | **64.79%** | 59.09% | **46.71%** | 14.47% |
> | 6  | 28 | 20 | 75.00%  | 45.00% | 18.42% | 13.16% |
> | 7  | 8  | 18 | 50.00%  | 38.89% | 5.26%  | 11.84% |
> | 8  | 1  | 15 | 100.00% | 40.00% | 0.66%  | 9.87% |
> | 9  | —  | 10 | —       | 40.00% | —      | 6.58% |
> | 10 | —  | 8  | —       | 37.50% | —      | 5.26% |
> | 12 | —  | 5  | —       | 40.00% | —      | 3.29% |
>
>
> ## Clintox Task:
>
> | Property Count | Attrilens-Mol Samples | Baseline Samples | Attrilens-Mol Acc. | Baseline Acc. | Attrilens-Mol Percentage | Baseline Percentage |
> |----------------|-------------------|------------------|----------------|--------------|----------------|--------------|
> | 1  | —  | 8  | —       | 50.00% | —      | 5.41% |
> | 2  | —  | 12 | —       | 25.00% | —      | 8.11% |
> | 3  | 36 | 18 | 88.89%  | 27.78% | 24.32% | 12.16% |
> | 4  | 25 | 22 | 88.00%  | 54.55% | 16.89% | 14.86% |
> | **5** | **64** | 28 | **93.75%** | 32.14% | **43.24%** | 18.92% |
> | 6  | 22 | 25 | 95.45%  | 24.00% | 14.86% | 16.89% |
> | 7  | —  | 15 | —       | 60.00% | —      | 10.14% |
> | 8  | —  | 10 | —       | 30.00% | —      | 6.76% |
> | 9  | 1  | 6  | 0.00%   | 33.33% | 0.68%  | 4.05% |
> | 10 | —  | 4  | —       | 25.00% | —      | 2.70% |
>
>
> As demonstrated in the tables above, the distribution of selected attribute undergoes significant changes after training, showing a clear preference for specific numbers of attributes across different tasks. For instance, the model tends to select 4/5/5 attributes in the BBBP/BACE/Clintox tasks, respectively. These findings indicate that our training strategy not only effectively guides the redistribution of attribute selection, but it also cultivates more discriminative selection preferences for specific tasks, thereby enhancing overall performance.

---

> ### Author Response · Authors · 2025-11-27
> **Response (Part 2)**
>
> We further include a quantitative analysis of the attribute distributions between Attrilens-Mol and baseline. As shown in the table below, we list the top-10 most frequently generated attributes along with their corresponding statistics. The numbers following each attribute indicate how many times the attribute was selected during the reasoning process. An upward arrow (↑) denotes a noticeable increase in selection frequency compared to the baseline model, while a downward arrow (↓) indicates a noticeable decrease.
>
> | Task (with Num. of Test Samples)          | Model                            | Common Attributes (with Frequency)                                                              | Distinctive Attributes (with Frequency)                                                                  |
> |:---------------|:---------------------------------|:------------------------------------------------------------------------------------------------|:---------------------------------------------------------------------------------------------------------|
> | BACE (#152)    | Attrilens-Mol                    | LogP (129) ↑, MolWeight (126), TPSA (115), HBD (76) ↓, HBA (63) ↓, Aromaticity (44)             | **AromaticRings (32) ↑, Charge (21) ↑, AmideGroups Count (19) ↑, ElectronWithdrawingGroups Count (8) ↑** |
> |                | Baseline LLM                     | MolWeight (121), TPSA (113), LogP (98), HBD (94), HBA (87), Aromaticity (39)                    | HOMO (56), LUMO (54), PolarSurfaceArea (45), HDonors (48)                                                |
> | BBBP (#204)    | Attrilens-Mol                    | MolWeight (196), LogP (195), TPSA (143), HBD (66) ↓, HBA (65) ↓, Charge (52) ↑, SurfaceArea (43) ↓ | **FunctionalGroups (22) ↑, PolarGroups (18) ↑, AromaticAtoms (12) ↑**                                    |
> |                | Baseline LLM                     | MolWeight (186), LogP (180), TPSA (135), HBD (91), SurfaceArea (86), HBA (80), Charge (36)      | PolarSurfaceArea (59), RingStrain (35), Aromaticity (21)                                                 |
> | ClinTox (#148) | Attrilens-Mol                    | MolWeight (143), LogP (142) ↑, TPSA (114), HBD (65) ↓, HBA (43) ↓, RingCount (47) ↓, FunctionalGroups (24) ↓ | **HydrogenCount (17) ↑, DegreeOfUnsaturation (12) ↑, RingStrain (7) ↑**                                  |
> |                | Baseline LLM                     | MolWeight (135), LogP (109), TPSA (105), HBD (92), HBA (81), RingCount (68), FunctionalGroups (54) | Electrophilicity (57), Aromaticity (49), PolarSurfaceArea (44)                                           |
>
> Specifically, our Attrilens-Mol maintains consistent identification of fundamental physicochemical attributes (e.g., MolWeight, LogP, TPSA, HBD, HBA) while dynamically optimizing the selection and weighting of task-specific attributes through our dual reward mechanism. The count reward ($R_{\text{count}}$) focuses the reasoning process by limiting attribute quantity, while the rational reward ($R_{\text{rational}}$) ensures selected attributes are chemically meaningful and predictive. This optimized attribute selection directly enhances feature discriminability, as evidenced by the improved decision tree performance as shown in Table 6, confirming that our method yields more interpretable and biologically relevant feature representations.
>
>
> > **W2.1**. Although the proposed reward design improves predictive performance, it still lags behind LLMs such as the GPT series and the DeepSeek series on some tasks.
>
> As shown in Table 2, AttriLens-Mol (i.e., R1-Distilled-LLaMA3.1-8B) achieves an average score **AVG$_4$** of 67.9 on multiple classification tasks (i.e., BBBP, BACE, ClinTox, SIDER), significantly surpassing DeepSeek-R1 (58.6) and GPT-4o (54.8) with **>100B** parameters. This demonstrates that AttriLens-Mol holds an advantage in overall classification accuracy across most scenarios, particularly in terms of parameter efficiency (using a model with only **8B** parameters). Our key contribution lies not in outperforming every single large model on every task, but in demonstrating an efficient approach through attribute-guided reinforcement learning, even small-scale models can exhibit significant potential in molecular property prediction.

---

> ### Author Response · Authors · 2025-11-27
> **Response (Part 3)**
>
> > **W2.2**. Moreover, there is no discussion or experimental comparison with the state-of-the-art model TxGemma, which is trained for molecular property tasks.
>
> We have carefully analyzed TxGemma [1] and wish to highlight a fundamental and methodological distinction as follows. (1) TxGemma establishes a strong foundation model through extensive supervised fine-tuning on large-scale bioinformatic data, while our work specifically investigates how reinforcement learning (RL) can enhance reasoning capabilities in molecular property prediction. This difference in objectives makes direct comparison challenging. (2) TxGemma aims to build a general-purpose chemistry model, whereas AttriLens-Mol focuses on optimizing interpretable reasoning through attribute-guided RL.
>
> Nevertheless, we show performance comparison in the table below. Specifically, TxGemma shows stronger performance on BBBP while our method demonstrates significant advantages on BACE (+14.3) and ClinTox (+7.2) tasks. This indicates that AttriLens-Mol provides particular benefits for complex molecular interaction prediction (BACE) and toxicity assessment (ClinTox), where structured reasoning about physicochemical properties is crucial. The performance pattern aligns with our focus on optimizing interpretable reasoning pathways rather than general foundation modeling.
>
>
> | Model              | #Training Samples |BBBP (↑)     | BACE (↑)    | ClinTox (↑) |
> |--------------------|-------------------|----------|----------|----------|
> | TxGemma-9B-Predict      | 7080k             | **66.7**     | 53.3 | 86.5 |
> | Attrilens-Mol (**Ours**) | 4k                |58.1 | **67.6**     |  **93.7**    |
>
>
>
> > **W3.1**. The instructions used for LLMs are not precisely described, limiting the reproducibility of this work.
>
> The complete instruction template used in our work has been added to the appendix D of the revised version.
>
> > **W3.2**. In addition, using broader datasets such as ToxCast or HIV would further strengthen the contribution of the proposed reward.
>
> Our dataset selection primarily follows the data curation strategies of LLaSMol [2] and AutoMolCo [3]. As shown in the table below, the results demonstrate that our method still achieves better results, which strongly reinforces the contribution of our proposed reward design, confirming its effectiveness in enhancing model generalization and robustness across diverse molecular property prediction tasks.
>
>
> | Model                                | HIV       | ToxCast （Androgen Receptor） |
> |--------------------------------------|-----------|---------------|
> | GPT-4o                               | 59.60     | 44.76         |
> | Qwen2.5-7B                           | 42.77     | 33.38         |
> | R1-Distilled-Qwen2.5-7B              | 41.21     | 31.59         |
> | R1-Distilled-Qwen2.5-7B (Ours, GRPO) | **59.69** | **46.93**     |
>
> > **Q2**. Are the individual reward components simply summed without weighting?
>
> Following the GRPO design described in DeepSeek-Math, we combine the reward components through direct summation, which also follows the standard princlple of reinforcement learning without specific weights for each sub-rewards. In practice, the performance is also not influenced by the weights.
>
>
> Reference:
> [1] Wang, Eric, et al. TxGemma: Efficient and Agentic LLMs for Therapeutics. arXiv 2504.06196.
> [2] Yu, Botao, et al. LlaSMol: Advancing large language models for chemistry with a large-scale, comprehensive, high-quality instruction tuning dataset. COLM 2024.
> [3] Zhang, Zimin, et al. Automated molecular concept generation and labeling with large language models. COLING 2025.
>
> ---
>
> Thank you again for your time and effort. We remain open to any further suggestions or requests you may have.

---

### Author Response · Authors · 2025-12-03
**General Response**

Dear all reviewers and AC,

We sincerely thank all reviewers for their thorough reviews and constructive feedback. We are encouraged that the reviewers recognize AttriLens-Mol's innovations in well-designed attribute-guided reward function, the significant performance improvements on OOD tasks with a smaller model (8B), and the overall solid methodology and clear presentation.

**Key Improvements and Clarifications**
1. Technical Novelty and Contribution Clarification
*   Through attribute-guided reinforcement learning, our method enabled smaller models (8B) to outperform much larger models (>100B) on benchmark datasets. This relies on the efficient reasoning of our method, rather than simply on model size and knowledge distillation. [**#i7Ns**]
*   Provided detailed comparison with recent work like TxGemma. [**#p6WF**]
*   Expanded discussion on the rationale behind using advanced LLMs for knowledge extraction. [**#anam**]

2. Extended Evaluation and Analysis
*   Added quantitative analysis of attribute distribution and prediction accuracy across tasks (BBBP, BACE, ClinTox). [**#p6WF**]
*   Included comparison experiments on additional benchmark datasets (HIV, ToxCast, Tox21). [**#p6WF**, **#anam**]
*   Provided analysis on the impact of the quality of guidance LLM. [**#i7Ns**]

3. Details Clarification and Reproducibility
*   Clarified the combination rule of reward components (direct sum operation following GRPO). [**#p6WF**]
*   Added the complete instruction template to the Appendix. [**#p6WF**]
*   Provided key hyperparameters and training settings, committing to full code release upon acceptance. [**#BsNe**]

We are informed that due to some uncontrollable reasons, interactions with the reviewers have become challenging and the scores cannot be altered. We regret this situation and hope that our response has addressed the existing concerns. We will ensure that the aforementioned improvements are incorporated into the final version to contribute to the advancement of this field.

Best Regards,
Authors

---

### Meta-Review · Area_Chair_fAsK · 2026-01-07

**Summary:**

This paper proposes an “attribute-guided RL” way to fine-tune a LLM for molecular property prediction. The reward is designed with task-agnostic rewards plus attribute-oriented rewards, including a count reward and a rationality reward ) Experiments on MoleculeNet-style tasks show the RL-tuned model improves over SFT/prompting baselines, with additional decision-tree analysis to argue the generated attributes are predictive features, not just plausible text.

Despite the fact that all reviewers agree that this paper's design is effective, the proposed methods heavily relies on a teacher LLMs as well as the RDKit descriptor. These dependence will significantly affect the generalizability of the proposed methods. Reviewers are also concerns about the justification of the reward heuristics and the empirical details.

**Reviewer Concerns:**

The authors provided a response justifying some of the reviewers' concern including the RL hyper parameter, empirical details, results on more datasets / benchmarks and the efficiency. However, several of the major concerns still remains open:
1. Regarding the reward model including the heuristics and the GPT evaluation, it's still not clear how this work. differs from the distillation and the RD-kit rule-based description seems very sparse. This is reflected in Reviewer i7Ns, BsNe, and other reviewers.
2. Reviewer anam questioned about the size of the dataset, this is not properly addressed in the response.

**Reviewer Scores:**

The reviewers provided a borderline rating as 4/4/6/6. However, to the AC's understanding, there are still fundamental challenges in the authors response which prevents the reviewers' in raising their score.

---

### Decision · Program_Chairs · 2026-01-26

Reject